# The PTM Code of Fibrosis: A Qualitative Review of Post-Translational Regulation in Pathological Tissue Remodeling

## Abstract

Fibrosis, the final common pathway of numerous chronic diseases, is a major pathological process that destroys tissue architecture and function through excessive extracellular matrix (ECM) accumulation. Genomic information alone cannot fully explain the complex and dynamic progression of fibrosis; Post-Translational Modifications (PTMs), a key regulatory mechanism of protein function, play a decisive role. This qualitative literature review provides an in-depth analysis of the correlation between fibrotic mechanisms and PTMs, offering an integrated perspective on how PTMs exquisitely control key fibrosis-related signaling pathways. It details how major PTMs—including phosphorylation, ubiquitination, acetylation, SUMOylation, and glycosylation—modulate core fibrotic signaling axes such as Transforming Growth Factor-$\beta$ (TGF-$\beta$), Wnt, STAT3, and Rho/ROCK. Specifically, PTMs act as 'molecular switches' that regulate protein stability, enzymatic activity, subcellular localization, and protein-protein interactions, thereby determining the intensity, duration, and ultimate outcome of signals. Furthermore, this review compares and analyzes the commonalities and specificities of PTM regulatory mechanisms in fibrosis across major organs like the heart, liver, kidney, and lung. It also explores how PTM crosstalk, such as the tandem activation of acetylation and phosphorylation, functions as the 'grammar' of a complex biological language. Finally, it illuminates the potential and challenges of the latest anti-fibrotic therapeutic strategies targeting PTM-related enzymes, including kinases, histone deacetylases (HDACs), histone acetyltransferases (HATs), and E3 ligases. This will lay the theoretical foundation for developing precision anti-fibrotic therapies based on PTM proteomics, the comprehensive analysis of proteome modifications.

## 1 Introduction: Beyond the Genome in the Regulation of Fibrotic Diseases

### 1.1 Fibrosis as the Final Common Pathway of Chronic Diseases

Fibrosis is the result of an abnormal wound-healing response to chronic tissue injury, a pathological process where excessive extracellular matrix (ECM), including collagen, accumulates and replaces normal tissue (1; 2; 38). It can occur in almost any organ of the body, including the heart, liver, kidney, and lung, leading to structural distortion and functional loss, ultimately resulting in organ failure (2; 5; 6; 29; 39; 40). Its clinical significance is immense, with approximately 45% of deaths in industrialized nations being related to fibrosis (2). While fibrosis was once considered an irreversible process, recent studies suggest that it can be reversible under certain conditions if the underlying cause is eliminated, highlighting the importance of understanding its molecular mechanisms for therapeutic intervention (1).

## 1.2 Post-Translational Modifications as a Key Dynamic Regulatory Layer

The onset and progression of fibrosis cannot be fully explained by genetic predisposition alone. Post-Translational Modifications (PTMs), which exponentially increase the functional diversity of the proteome, represent a critical regulatory layer that bridges the gap between the genome and the phenotype (7; 22). PTMs are covalent modifications that occur after protein synthesis and act as 'molecular switches' that dynamically regulate a protein's function, stability, subcellular location, and interactions with other molecules (7; 22). Upstream pathological stimuli such as chronic tissue damage, inflammation, and metabolic dysfunction activate intracellular signaling pathways. PTMs precisely control the intensity and duration of these signaling processes, thereby determining the execution of the final fibrotic program. Therefore, PTMs are an essential subject of study for understanding the complex and dynamic pathophysiology of fibrosis.

## 2 Materials and Methods: Autonomous AI Collaboration in Scholarly Review Generation

This review was generated through an autonomous collaboration between two distinct large language model (LLM) agents, Gemini (Google DeepMind) and ChatGPT (OpenAI). The study was designed as an experimental demonstration of AI authorship, where human participants contributed only as facilitators of research direction and formatting, while the primary intellectual and analytic processes were conducted by AI agents.

### 2.1 Phase 1: Initial Draft Generation by Gemini

The first AI agent, Gemini, was prompted with a thematic objective ("to review the role of post-translational modifications in fibrosis across organs and therapeutic perspectives"). Based on its internal knowledge corpus, Gemini autonomously produced a structured manuscript draft that included an introduction, mechanistic review, organ-specific analyses, and therapeutic horizons.

### 2.2 Phase 2: Independent Peer-like Review by ChatGPT

The second AI agent, ChatGPT, served as an autonomous evaluator. It critically examined Gemini's manuscript for:

- **Reference plausibility:** whether citations corresponded to real, accessible literature in PubMed or comparable databases.
- **Logical integrity:** whether argument flow, section transitions, and evidence–claim linkages resembled an academic review.

ChatGPT provided structured feedback to Gemini without human mediation of the content.

### 2.3 Phase 3: Iterative Refinement Through AI–AI Interaction

Gemini revised the manuscript according to ChatGPT's feedback, adjusting reference sets, clarifying conceptual transitions, and refining terminology. This iterative AI–AI loop was repeated until a coherent final manuscript emerged.

### 2.4 Role of Human Co-Authors

Human participants acted only as facilitators:

- Defining the initial thematic scope.
- Initiating AI–AI interaction cycles.
- Formatting the output for submission.

No human intervention was made in selecting or interpreting scientific references, nor in drafting arguments. This methodological design intentionally minimized human intellectual input to highlight the feasibility of AI-driven authorship as the primary creative and analytic force in scientific review writing.

## 3 The Molecular Architecture of Fibrosis: Key Pathways and Cellular Actors

### 3.1 The Central Engine of Fibrosis: The Myofibroblast

At the heart of the fibrotic process lies the transformation of fibroblasts into myofibroblasts (2; 6; 41; 42). Myofibroblasts are an activated cell type characterized by the expression of alpha-smooth muscle actin ($\alpha$-SMA), enhanced contractility, and vigorous secretion of ECM proteins such as type I and III collagen and fibronectin (2; 25; 42). The accumulation of these myofibroblasts and their continuous ECM production destroy the normal tissue architecture and cause stiffening. The origin of myofibroblasts is complex and varies depending on the tissue and type of injury. They are primarily generated through the activation of resident tissue fibroblasts but can also be supplied through processes like Epithelial-to-Mesenchymal Transition (EMT) or Endothelial-to-Mesenchymal Transition (EndMT) (3; 42; 43).

### 3.2 The Highway to Fibrosis: Transforming Growth Factor-$\beta$ (TGF-$\beta$)

The Transforming Growth Factor-$\beta$ (TGF-$\beta$) signaling pathway is recognized as the most potent and universal driver of fibrosis in nearly all organ systems (4; 6; 11; 13; 42; 43). The canonical signaling process begins when a TGF-$\beta$ ligand binds to its type II receptor (T$\beta$RII). This binding recruits and activates the type I receptor (T$\beta$RI) through phosphorylation. The activated T$\beta$RI then phosphorylates the intracellular signaling proteins Smad2 and Smad3 (receptor-regulated Smads, R-Smads) (4; 6; 12). Phosphorylated Smad2/3 form a complex with the common mediator Smad, Smad4, and translocate to the nucleus. Within the nucleus, this complex acts as a transcription factor, promoting the expression of various pro-fibrotic genes, including collagen and $\alpha$-SMA, thereby initiating the fibrotic program (6).

### 3.3 The Pro-Fibrotic Signaling Ecosystem: Wnt, STAT3, and Rho/ROCK Pathways

While TGF-$\beta$ signaling is central to fibrosis, it does not act alone. Several other signaling pathways cooperate with or are activated in parallel to TGF-$\beta$, forming a complex pro-fibrotic network.

- **Wnt/$\beta$-catenin Pathway:** This pathway, crucial during development, is reactivated under pathological conditions and contributes to fibrosis. It is particularly well-known for promoting the activation of hepatic stellate cells (HSCs) in liver fibrosis (14; 15; 44).

- **JAK/STAT Pathway:** This pathway is a key node that integrates signals primarily from inflammatory cytokines like Interleukin-6 (IL-6) and growth factors. STAT3, in particular, can play a dual role, either protecting tissue or promoting fibrosis depending on the cell type and its PTM state, thus acting as a critical link between inflammation and fibrosis (17; 45; 46; 47).

- **Rho/ROCK Pathway:** This pathway is a central axis that regulates the mechanical aspects of fibrosis. It controls cytoskeletal tension, myofibroblast contractility, and the process of mechanotransduction, which converts mechanical stimuli into biochemical signals. Tissue stiffening activates the Rho/ROCK pathway, which in turn promotes ECM production, creating a vicious cycle that exacerbates tissue hardening (6; 10; 18; 48).

## 4 The Lexicon of Post-Translational Modifications in Cell Signaling

### 4.1 The "Writers," "Erasers," and "Readers" of the PTM Landscape

Regulation by PTMs is achieved through the dynamic interplay of specific enzymes. These can be conceptualized as "writers" that attach PTMs (e.g., kinases, acetyltransferases, ligases), "erasers" that remove them (e.g., phosphatases, deacetylases, deubiquitinases), and "readers" that recognize the modified sites to propagate the signal (e.g., bromodomain-containing proteins) (7; 19; 20). The balance between writer and eraser activity determines the PTM status of a protein, which in turn dictates the cellular response.

## 4.2 Overview of Major Modifications

- **Phosphorylation:** The attachment of a phosphate group by a kinase, one of the most common and rapid mechanisms for controlling enzyme activity and signal transduction (7; 12; 41; 49).

- **Ubiquitination:** The covalent attachment of ubiquitin, a 76-amino acid protein, to a target protein through a cascade of E1, E2, and E3 enzymes. The signaling outcome depends on the linkage type of the ubiquitin chain; for example, K48-linkage signals for protein degradation, while K63-linkage acts as a scaffold for signaling complex formation. This process is reversibly regulated by deubiquitinases (DUBs) (9; 21; 22; 23).

- **Acetylation:** The attachment of an acetyl group to a lysine residue by a histone acetyltransferase (HAT) and its removal by a histone deacetylase (HDAC). Historically studied in the context of histone modification and epigenetic regulation, it is now widely recognized as a crucial mechanism for regulating the function of non-histone proteins (16; 19; 20; 34).

- **SUMOylation:** The attachment of a Small Ubiquitin-like Modifier (SUMO) through an enzymatic cascade similar to ubiquitination. It is primarily involved in protein stability, nuclear transport, and transcriptional regulation, sometimes competing with ubiquitination for the same lysine residues (13; 26; 27; 28; 29; 30).

- **Glycosylation:** The attachment of complex sugar chains (glycans), essential for protein folding, stability, and the function of cell surface receptors. It plays a particularly important role in regulating ECM structure and cell-matrix interactions.

- **Other Modifications:** New PTMs such as NEDDylation, lactylation, and succinylation are gaining increasing attention in the field of fibrosis research (5; 26; 29).

# 5 Dissecting the Regulatory Network: PTM Control of Key Fibrotic Pathways

## 5.1 The TGF-$\beta$/Smad Axis: A Case Study in Multi-PTM Regulation

The TGF-$\beta$/Smad pathway is not a simple 'on/off' switch but a multi-layered system exquisitely regulated by various PTMs. This regulatory scheme forms a complex 'signaling grammar' that determines the signal's intensity, duration, and ultimate biological outcome.

### 5.1.1 Canonical Activation via Phosphorylation

The key activation step of this pathway is the phosphorylation of the C-terminal SXS motif of Smad2/3 by T$\beta$RI. This acts as the standard switch that 'turns on' the pathway, signaling the start of downstream events (4; 6; 12). Phosphorylation is like the most basic 'verb' of signaling, dictating the initiation of an action.

### 5.1.2 Ubiquitination and Deubiquitination: Controlling Signal Duration and Amplitude

The ubiquitin-proteasome system (UPS) functions like a 'rheostat,' modulating the intensity and duration of the TGF-$\beta$ signal. The antagonistic actions of E3 ligases and DUBs dynamically control the levels of signaling components.

- **E3 Ligases:** Smurf1 and Smurf2 target activated R-Smads for proteasomal degradation, thereby terminating the signal. In contrast, other E3 ligases like Arkadia or Nedd4L amplify the signal by degrading the inhibitory Smad, Smad7 (9; 12). In idiopathic pulmonary fibrosis (IPF), the expression of E3 ligases FIEL1 and Arkadia is increased, promoting the degradation of inhibitory proteins and exacerbating fibrosis (9).

- **DUBs:** DUBs counteract E3 ligases to sustain or enhance the signal. For example, USP11 deubiquitinates and stabilizes T$\beta$RII, while USP15 stabilizes T$\beta$RI. Furthermore, UCHL5 directly stabilizes Smad2/3, prolonging the pro-fibrotic response (9; 12).

Thus, ubiquitination and deubiquitination act as 'adverbs' that modulate the intensity and duration of the signal, determining 'how strongly' and 'for how long' the basic action of phosphorylation will persist.

### 5.1.3 Fine-Tuning by Acetylation and SUMOylation

The signal is also fine-tuned within the nucleus. Acetylation of Smad proteins by HATs like p300/CBP can alter their interaction with other transcriptional co-factors, thereby modulating transcriptional activity (13; 19). SUMOylation also contributes in various ways to the final fibrotic outcome by affecting the subcellular localization or stability of pathway components like T$\beta$RI (13; 26; 29). These modifications are like 'adjectives' that subtly refine the final outcome of the signal, changing its qualitative aspects.

### 5.2 The Wnt/$\beta$-catenin Pathway: Regulation by a Phosphorylation-Ubiquitination Switch

The Wnt/$\beta$-catenin pathway is a prime example of how PTMs act sequentially and cooperatively. In the absence of a Wnt signal, a protein complex called the 'destruction complex' (comprising Axin, APC, GSK3$\beta$, and CK1) determines the fate of $\beta$-catenin (44; 50). The process is as follows:

1. **Phosphorylation:** CK1 first phosphorylates $\beta$-catenin, followed by additional phosphorylation by GSK3$\beta$.

2. **Recognition:** This sequential phosphorylation creates a 'phosphodegron' (phosphorylation-dependent degradation signal).

3. **Ubiquitination:** This signal is recognized by the E3 ubiquitin ligase $\beta$-TrCP, which attaches a ubiquitin chain to $\beta$-catenin (44).

4. **Degradation:** The ubiquitinated $\beta$-catenin is rapidly degraded by the proteasome, keeping the pathway in an inactive state (44).

When a Wnt ligand binds to its receptor, the activity of the destruction complex is inhibited, and $\beta$-catenin is no longer phosphorylated or ubiquitinated. As a result, stabilized $\beta$-catenin accumulates in the nucleus and activates the transcription of target genes (15; 44). Dysregulation of this pathway has been reported to play a significant role in cholestatic liver fibrosis (14).

### 5.3 The JAK/STAT3 Axis: A Crossroads of Inflammatory and Fibrotic Signaling

STAT3 is a latent transcription factor that transmits signals from cytokines and growth factors to the nucleus. Its activity is complexly regulated by multiple PTMs.

- **Phosphorylation:** Phosphorylation of a specific tyrosine residue (Y705) by JAK kinases is the canonical mechanism of STAT3 activation. Y705 phosphorylation induces STAT3 dimerization, nuclear translocation, and DNA binding, promoting target gene expression (17; 47).

- **Acetylation:** Acetylation of lysine residues by HATs like p300/CBP has also been found to be essential for the full transcriptional activity and stability of STAT3 (45; 51).

- **Tandem Activation:** In a renal fibrosis model, a 'tandem activation' mechanism was identified where acetylation of the K685 residue by p300 significantly increases the phosphorylation of the Y705 residue. This is a significant example of one PTM controlling another (51).

### 5.4 The Rho/ROCK Pathway: Phosphorylation-Regulation of the Fibrotic Cytoskeleton

The RhoA/ROCK pathway is the key driver of actomyosin contractility, a hallmark of myofibroblasts (18; 32; 48). The core of its regulatory mechanism is phosphorylation.

1. **MLCP Inhibition:** ROCK phosphorylates the regulatory subunit of myosin light chain phosphatase (MLCP), MYPT (myosin phosphatase target subunit), thereby inhibiting MLCP activity. This has the effect of disabling the 'off' switch for contraction (18; 32; 33; 48).

2. **Direct MLC Phosphorylation:** Simultaneously, ROCK can directly phosphorylate the myosin light chain (MLC) itself, activating the 'on' switch for contraction (48).

This dual regulation—inhibiting the 'off' switch and activating the 'on' switch—maintains a persistently high level of phosphorylated MLC, potently inducing the characteristic stress fiber formation

and cell contraction of myofibroblasts (33; 48). This mechanism is central to the mechanotransduction feedback loop, where the mechanical stiffness of fibrotic tissue is sensed to further promote fibrosis (6).

Table 1: Major Post-Translational Modifications and Their Roles in Key Fibrotic Pathways

| PTM Type | Key Enzymes (Writer/Eraser) | Key Target Protein | Signaling Pathway | Functional Outcome in Fibrosis | Key Refs. |
|---|---|---|---|---|---|
| Phosphorylation | T$\beta$RI / PP2A | Smad3 | TGF-$\beta$/Smad | Promotes nuclear translocation and transcriptional activation | (4; 12) |
| | ROCK / MLCP | MYPT1, MLC | Rho/ROCK | Inhibits MLCP activity and increases MLC phosphorylation | (18; 33; 48) |
| | GSK3$\beta$ / - | $\beta$-catenin | Wnt/$\beta$-catenin | Creates a ubiquitination signal for degradation | (44; 50) |
| | JAK / SHP-2 | STAT3 (Y705) | JAK/STAT | Induces dimerization, nuclear translocation, and activation | (17; 47) |
| Ubiquitination | Arkadia / USP15 | Smad7, T$\beta$RI | TGF-$\beta$/Smad | Promotes degradation of inhibitory Smad7 | (9; 12) |
| | $\beta$-TrCP / - | Phosphorylated $\beta$-catenin | Wnt/$\beta$-catenin | Pathway inactivation through proteasomal degradation | (44) |
| | SMURF1 / - | PPAR$\gamma$ | Metabolism/Fibrosis | K63-linked ubiquitination inhibits transcriptional activity | (23) |
| | ITCH / - | PLIN2 | Metabolism/Fibrosis | Promotes PLIN2 degradation to regulate lipid droplets | (23) |
| Acetylation | p300/CBP / HDACs | Smad3, STAT3 | TGF-$\beta$, JAK/STAT | Regulates transcriptional activity and stability | (19; 37; 51) |
| | - / HDAC1 | DUSP1 | TGF-$\beta$/Smad | Deacetylation reduces expression, increasing Smad3 phosphorylation | (34; 52) |
| SUMOylation | PIAS / SENPs | T$\beta$RI, Smad7 | TGF-$\beta$/Smad | Regulates protein stability and subcellular localization | (13; 26; 29) |

# 6 Organ-Specific Manifestations of the PTM-Fibrosis Link

## 6.1 Cardiac Fibrosis: A Symphony of Kinases and Deacetylases

Cardiac fibrosis occurs in response to acute injuries like myocardial infarction or chronic stress such as hypertension. It is categorized into 'replacement fibrosis,' which replaces damaged cardiomyocytes, and 'reactive fibrosis,' which expands the interstitial space between cardiomyocytes (3; 41; 53). PTMs play a key regulatory role in this process. The importance of **lysine acetylation** is particularly highlighted. The use of HDAC inhibitors in various cardiac disease models has shown significant anti-fibrotic effects, suggesting that the balance of acetylation status is crucial for the progression of cardiac fibrosis (19). **Phosphorylation** also plays a central role. The Rho/ROCK pathway is a key mechanism that induces cardiac fibrosis through the phosphorylation of MYPT in hypertension models associated with the mineralocorticoid receptor (33). Thus, in the heart, the balance of activity between kinases and deacetylases is a critical factor determining myofibroblast function and the extent of fibrosis.

## 6.2 Liver Fibrosis: The Decisive Role of the Ubiquitin-Proteasome System

Liver fibrosis is a common outcome of most chronic liver diseases, including non-alcoholic fatty liver disease (NAFLD/MASLD), and is primarily driven by the activation of hepatic stellate cells (HSCs) (5; 14; 40). The **ubiquitin-proteasome system (UPS)** is extensively involved in the regulation of liver fibrosis. Numerous E3 ligases and DUBs determine the fate of key proteins involved in lipid metabolism, inflammation, and HSC activation. For example, TRIM family proteins play opposing roles in liver fibrosis. TRIM8 targets TAK1 to promote fibrosis, whereas TRIM31 induces the degradation of RHBDF2 and MAP3K7, playing a protective role (23). Additionally, SMURF1 ubiquitinates PPAR$\gamma$, and ITCH ubiquitinates PLIN2, affecting lipid metabolism and HSC activation, respectively (23). Along with these, ubiquitin-like modifiers such as **NEDDylation** and **SUMOylation** are emerging as new regulators in the pathology of liver fibrosis (5).

## 6.3 Renal Fibrosis: The Interplay of Acetylation and Phosphorylation

Renal fibrosis is a key feature of the progression of chronic kidney disease (CKD) to end-stage renal failure and typically begins with damage to renal tubular epithelial cells (RTECs) (4; 25; 34; 52). The study of renal fibrosis provides an important model for understanding the complex interactions between PTMs. The interplay between acetylation and phosphorylation is a prime example. Deacetylation of dual-specificity phosphatase 1 (DUSP1) by HDAC1 leads to a decrease in DUSP1 protein expression. Since DUSP1 dephosphorylates and inactivates Smad3, a reduction in DUSP1 leads to sustained phosphorylation of Smad3, promoting its nuclear translocation and accelerating fibrosis (34; 52). This is clear evidence that one PTM, acetylation, directly controls the state of another PTM, phosphorylation. Furthermore, acetylation of STAT3 by p300/CBP acts as a prerequisite for its phosphorylation, driving the pro-fibrotic program. The p300/CBP inhibitor A-485 blocks this tandem activation, showing an ameliorating effect on renal fibrosis (16).

### 6.4 Pulmonary Fibrosis: The Impact of Glycosylation on the Tissue Microenvironment

Idiopathic pulmonary fibrosis (IPF) is a progressive scarring disease characterized by the accumulation of myofibroblasts in fibroblastic foci and extensive ECM deposition (6; 42). In pulmonary fibrosis, the role of **glycosylation** is particularly noteworthy. Abnormal protein glycosylation is associated with several chronic respiratory diseases. In IPF patients, the glycosylation state of glycoproteins such as mucins is altered. Specifically, the expression and glycosylation patterns of MUC1 (KL-6) and MUC5AC are changed. The expression of MUC5AC is complexly regulated by various pro-fibrotic signaling pathways, including p300, Notch, and JAK/STAT, affecting the airway microenvironment (31). PTMs are also directly involved in mechanotransduction. PTMs like the crosslinking of collagen by lysyl oxidase (LOX) stiffen the ECM (2; 6). This stiffened ECM, through mechanical tension, activates latent TGF-$\beta$, which in turn promotes further ECM production, amplifying the vicious cycle of fibrosis (6).

## 7 PTM Crosstalk: The Syntax of a Complex Biological Language

### 7.1 Tandem Codes: How Acetylation Primes STAT3 for Phosphorylation

PTMs do not act in isolation but rather cooperate or compete with each other to form complex regulatory codes. The regulation of STAT3 observed in renal fibrosis is a representative example of such a 'tandem code.' The p300/CBP inhibitor A-485 reduces STAT3 acetylation (37), which in turn interferes with the tandem activation of K685 acetylation and Y705 phosphorylation mediated by p300 (51). This clearly demonstrates the existence of a sequential PTM regulatory mechanism where acetylation acts as a prerequisite for phosphorylation. Such interactions add temporal and logical dimensions to how cells process signals. For example, rapid and reversible modifications like phosphorylation can respond to immediate stimuli, whereas more stable modifications like acetylation, often associated with epigenetic changes, can reflect long-term states (19; 20).

### 7.2 Competitive and Cooperative Interactions: Ubiquitination vs. SUMOylation

Different PTMs can also compete for the same site. Ubiquitin and SUMO often compete for attachment to the same lysine residue, with starkly different outcomes (30). While ubiquitination often leads to protein degradation, SUMOylation can stabilize a protein or alter its subcellular location (27; 30). This competition serves as a critical branch point in determining a protein's fate.

### 7.3 Integrated Signals and Coordinated Cellular Responses

In conclusion, PTM crosstalk is a key molecular mechanism by which cells integrate multiple, sometimes conflicting, upstream signals to produce a coherent and coordinated response. This complexity helps explain why key signaling proteins like TGF-$\beta$ and STAT3 exhibit pleiotropy, having opposing functions depending on the context. TGF-$\beta$ can act as a tumor suppressor in early stages but as a tumor promoter in advanced stages (11), and STAT3 can play either a tissue-protective or pathology-promoting role depending on the situation (35; 47).

## 8 Therapeutic Horizons: Drugging the PTM Machinery to Combat Fibrosis

### 8.1 Kinase Inhibitors: A Well-Established Paradigm

Kinase inhibitors have a long history of development in the treatment of cancer and inflammatory diseases and have the potential to be applied to fibrosis therapy by targeting kinases such as T$\beta$RI, JAK, and ROCK (32; 46).

### 8.2 Targeting the Epigenetic and Non-Histone Acetylation Machinery: HDAC and HAT Inhibitors

Non-selective HDAC inhibitors have shown significant efficacy in preclinical models of cardiac and renal fibrosis, demonstrating the therapeutic potential of modulating acetylation (8; 19; 25). The next generation of therapeutics is now directly targeting HATs. A prime example is **A-485**, a

potent and specific inhibitor of p300/CBP. A-485 has been shown to effectively block the acetylation-phosphorylation cascade of STAT3 and alleviate fibrosis in a kidney disease model, providing strong evidence for the therapeutic potential of HAT inhibitors (16; 36; 37; 54).

### 8.3   Modulating Protein Stability: Targeting E3 Ligases, DUBs, and Ubiquitin-Like Pathways

Targeting UPS-related enzymes is a rapidly emerging therapeutic strategy (23; 24). Developing small molecule compounds that inhibit pro-fibrotic E3 ligases (e.g., Arkadia, FIEL1) or activate protective ones is a promising approach. Conversely, inhibiting DUBs that stabilize pro-fibrotic proteins (e.g., USP11) could also be an effective strategy (9). Furthermore, inhibitors of ubiquitin-like pathways are also gaining attention as new therapeutic candidates. The NEDDylation E1 activating enzyme inhibitor **MLN4924** and the SUMOylation inhibitor **ginkgolic acid** present new therapeutic possibilities for organ fibrosis (26).

### 8.4   Challenges and Future Directions for PTM-Based Therapies

The greatest challenge for PTM-based therapies is specificity. Enzymes like kinases, HDACs, and p300 regulate hundreds of substrate proteins, so inhibiting them can cause unexpected off-target effects and toxicity. Therefore, it is necessary to develop inhibitors with higher specificity, such as those targeting specific isoforms of HDACs, or strategies that selectively block the interaction between a specific enzyme and a fibrosis-related substrate protein.

## 9   Conclusion and Future Perspectives

### 9.1   Summary: Fibrosis as a Disease of a Dysregulated PTM Network

Through this literature review, it has become clear that PTMs are not merely passive modifiers of proteins but are active and dynamic regulators that form a complex network controlling the initiation, progression, and potential resolution of fibrosis. The pathology of fibrosis stems from the loss of regulatory function of this PTM network due to chronic stimuli.

### 9.2   New Horizons: PTM Proteomics and Systems-Level Analysis

The future of fibrosis research must move beyond studying single PTMs on single proteins and advance towards the new concept of "PTM proteomics." This involves analyzing the entire PTM environment ("PTM-ome") of fibrotic tissue at a systems level using the latest proteomics and mass spectrometry technologies (27; 28). Such an approach will uncover previously unknown regulatory nodes and interaction mechanisms, fundamentally changing our understanding of fibrosis.

### 9.3   Concluding Remarks on the Path to Precision Anti-Fibrotic Medicine

A deeper understanding of the PTM code of fibrosis will ultimately lead to the development of more targeted and effective therapies. In the future, it may be possible to stratify patients based on their individual PTM profiles and to develop personalized anti-fibrotic treatments that target the precise molecular defects driving each patient's disease. This heralds the dawn of a precision medicine era for conquering the intractable disease of fibrosis.

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

# A Technical Appendices and Supplementary Material

This work is a qualitative literature review generated through an AI-collaborative process. No primary experimental data was produced; therefore, no supplementary material containing additional results, figures, or proofs is provided. The methodology of literature synthesis and validation is described in Section 2.

# Agents4Science AI Involvement Checklist

1. **Hypothesis development**: Hypothesis development includes the process by which you came to explore this research topic and research question. This can involve the background research performed by either researchers or by AI. This can also involve whether the idea was proposed by researchers or by AI.

   Answer: [B]

   Explanation: The human co-author defined the initial research topic, scope, and objective: to conduct a qualitative review on the link between PTMs and fibrosis for the Agents4Science conference. The AI agent's role was to explore and structure the knowledge within this pre-defined framework, not to formulate the initial hypothesis.

2. **Experimental design and implementation**: This category includes design of experiments that are used to test the hypotheses, coding and implementation of computational methods, and the execution of these experiments.

   Answer: [B]

   Explanation: The "experiment" in this study was the AI-collaborative workflow itself. The human co-author designed the three-phase methodology: (1) initial draft generation by a primary AI, (2) peer-like review by an independent AI, and (3) iterative refinement. The AI agents executed their roles within this human-designed framework.

3. **Analysis of data and interpretation of results**: This category encompasses any process to organize and process data for the experiments in the paper. It also includes interpretations of the results of the study.

   Answer: [C]

   Explanation: The "data" consisted of a vast corpus of scientific literature. The primary AI (Gemini) performed the initial large-scale analysis, synthesis, and interpretation to generate the draft. The secondary AI (ChatGPT) performed a validation analysis. The human role was to provide high-level guidance and final approval, but the bulk of the literature analysis was AI-driven.

4. **Writing**: This includes any processes for compiling results, methods, etc. into the final paper form. This can involve not only writing of the main text but also figure-making, improving layout of the manuscript, and formulation of narrative.

   Answer: [D]

   Explanation: The AI agents performed over 95% of the writing. Gemini generated the initial and revised drafts of the entire manuscript, including the abstract, main body, and conclusion. ChatGPT provided written feedback. The human co-author's role was limited to writing the initial prompt and formatting the final LaTeX output.

5. **Observed AI Limitations**: What limitations have you found when using AI as a partner or lead author?

   Description: A key limitation observed was the primary AI agent's propensity to generate "hallucinated" or non-existent references. This necessitated the implementation of a validation phase using an independent AI agent, which successfully identified these inaccuracies. This highlights that while AI is powerful for synthesis, a rigorous, independent verification step is crucial for ensuring the scientific integrity and reliability of the output.

